# Genetic InfoMax: Exploring Mutual Information Maximization in High-Dimensional Imaging Genetics Studies

**Yaochen Xie** *ethanycx@tamu.edu*
*Texas A&M University*

**Yuchao Lin** *kruskallin@tamu.edu*
*Texas A&M University*

**Ziqian Xie** *ziqian.xie@uth.tmc.edu*
*University of Texas Health Science Center*

**Sheikh Muhammad Saiful Islam** *sheikh.muhammad.saiful.islam@uth.tmc.edu*
*University of Texas Health Science Center*

**Degui Zhi** *degui.zhi@uth.tmc.edu*
*University of Texas Health Science Center*

**Shuiwang Ji** *sji@tamu.edu*
*Texas A&M University*

**Reviewed on OpenReview:** *https://openreview.net/forum?id=9UgUMFW67X*

## Abstract

Genome-wide association studies (GWAS) are used to identify relationships between genetic variations and specific traits. When applied to high-dimensional medical imaging data, a key step is to extract lower-dimensional, yet informative representations of the data as traits. Representation learning for imaging genetics is largely under-explored due to the unique challenges posed by GWAS in comparison to typical visual representation learning. In this study, we tackle this problem from the mutual information (MI) perspective by identifying key limitations of existing methods. We introduce a trans-modal learning framework Genetic InfoMax (GIM), including a regularized MI estimator and a novel genetics-informed transformer to address the specific challenges of GWAS. We evaluate GIM on human brain 3D MRI data and establish standardized evaluation protocols to compare it to existing approaches. Our results demonstrate the effectiveness of GIM and a significantly improved performance on GWAS.

## 1 Introduction

Genome-wide association studies (GWAS) have been an effective approach driving genetic discovery in the past 15 years (Abdellaoui et al., 2023). Given a phenotype of interest and a cohort of individuals with both the measurements of the phenotype and the genotypes over markers across the genome, linear or linear mixed models are built to test for the association of each marker to the phenotype and thus pinpoint the gene loci relevant to the phenotype. While the typical GWAS studies are focused on well-established phenotypes, typically the risks of diseases, or well-established macro-level measurements such as heights, and BMI, or molecular-level measurements such as protein and metabolomic biomarkers, there is an interest in expanding the scope of GWAS to focus on phenotypes that are derived from high-dimensional complex data modality such as medical imaging data, to better understand the biology of the underlying images. Such derived phenotypes are sometimes called endophenotypes as they serve as an intermediate between genetics and some

diseases. For example, brain images are often considered endophenotypes of brain-related diseases such as Alzheimer's disease. However, there is a lack of sophisticated approaches for deriving phenotypes for GWAS. Again, taking brain imaging as an example, existing approaches mostly used traditional non-learning software to derive brain region-based volumetric or surface features. Such methods, while valuable, are reaching a limit in fully capturing phenotype complexity and can narrow the scope of genetic discoveries. In this context, the advent of deep learning techniques offers a promising avenue for expanding the toolkit available for GWAS.

Recently deep learning approaches (Patel et al., 2022; Xie et al., 2022; Kirchler et al., 2022; Taleb et al., 2022) derive phenotypes from medical images by learning a latent representation that captures the inherent content of the input image. However, approaches learning from imaging data alone fail to utilize the accompanying genetic data. Those approaches tend to capture patterns that are not related to genes and common patterns shared by multiple individuals. For example, Patel et al. (2022) found that representations learned by an image autoencoder are unable to fully reconstruct fine details that are individually specific. To overcome these limitations, a solution is to incorporate trans-modal learning strategies that utilize the pairwise relationship between imaging and genetic data, such as trans-modal contrastive learning (Liang et al., 2022; Zolfaghari et al., 2021; Taleb et al., 2022). Unfortunately, the use of genetic data, including the encoding of data and capturing image-genetic relationships, still poses significant challenges. Results by Taleb et al. (2022) suggest that, despite the promising results for downstream classification of disease risk, multi-modal contrastive approaches still underperform compared to typical image-only approaches (Xie et al., 2022) on GWAS tasks. This underperformance becomes even more pronounced for higher-dimensional 3D data.

In this work, we formulate the problem as learning the representation of imaging data that shares the maximum mutual information with genetic data. By using mutual information as a perspective, we are able to examine the key reasons for the failure of typical trans-modal contrastive learning for GWAS on high-dimensional imaging data. To push the limits of existing learning approaches, we propose **Genetic InfoMax** (**GIM**), a trans-modal learning framework that includes a regularized mutual information estimator and a novel transformer-based genetic encoder. The framework addresses the issues of dimensional collapse and non-generalizable associations in representation learning for GWAS and fully utilizes the genetic data with physical and genetic position information. Our experiments demonstrate that GIM significantly improves performance on all four evaluation metrics.

## 2   Problem Formulation

We study the problem of genome-wide association studies (GWAS) on high-dimensional data. The GWAS aims to identify associations between specific genetic variants, known as single nucleotide polymorphisms (SNPs), in the genome and certain traits of interest such as the risk of disease and other biological characteristics of organisms. In particular, each individual genetic data is denoted by $\boldsymbol{G} = \{\boldsymbol{g}_1, \cdots, \boldsymbol{g}_L\}$, and traits of interest denoted by $\boldsymbol{y}$. The GWAS process involves statistical tests on the sample pairs $\{(\boldsymbol{G}_i, \boldsymbol{y}_i)\}_{i=1,\cdots,M}$ from a large number of $M$ individuals to identify the specific subset $\boldsymbol{G}^{g \to y} \subset \boldsymbol{G}$ of SNPs that are associated with the target traits $\boldsymbol{y}$. Here $L$ is the number of SNPs and each $\boldsymbol{g}_i \in \{0, 1, 2\}$ represents the number of carried variants for each individual. The values of nearby SNPs are often correlated due to their common inheritance from a shared ancestor. To account for this, it is necessary to select an independent subset of genetic information $\boldsymbol{G}^{\text{ind}} \subset \boldsymbol{G}^{g \to y}$, which can be achieved by clustering and selecting the SNPs with the lowest p-value from each cluster after the statistical test, which is important for accurate analysis and interpretation of the genetic data. Practically, when conducting GWAS on high-dimensional data $\boldsymbol{Y}$ such as medical imaging, an additional step is required before performing statistical tests. This step involves reducing the number of traits from the high-dimensional data $\boldsymbol{Y}$ to a smaller number $\boldsymbol{y}$ through experts' diagnosis or computational approaches.

**Traits Computing as Representation Learning.** To enable GWAS on high-dimensional data, we are interested in computationally obtaining informative lower-dimensional traits, termed GWAS representation learning, from the high-dimensional data. Specifically, for any high-dimensional data $\boldsymbol{Y}$ to be studied, the problem is formulated as learning lower-dimensional representations of $\boldsymbol{Y}$ with a corresponding encoder $f_\theta$ in a self-supervised manner, such that a larger number of independent SNPs $|G^{\text{ind}}|$ can be identified from the pairs $\{(\boldsymbol{G}_i, f_\theta(\boldsymbol{Y}_i))\}_{i=1,\cdots,M}$. The goal of identifying more independent SNPs requires that more information related to genetic variants is captured by the representation of $\boldsymbol{Y}$. We focus on learning $d$-dimensional

representations $\boldsymbol{y} = f_\theta(\boldsymbol{Y}) \in \mathbb{R}^q$ with the following optimization objective

$$\theta^* = \arg\max_\theta \mathcal{I}(f_\theta(\boldsymbol{Y}), \boldsymbol{G}), \tag{1}$$

where $\mathcal{I}(\cdot, \cdot)$ denotes the mutual information (MI) between two random variables. As computing the true value of mutual information is intractable, it becomes critical to develop an appropriate mutual information estimation under the GWAS problem setting.

**Notations of Data.** We instantiate our problem specifically with the 3D human brain magnetic resonance imaging (MRI) data and SNPs from the human genome. We denote the 3D brain MRI by $\boldsymbol{Y} \in \mathbb{R}^{H \times W \times D \times 1}$, where $H$, $W$, $D$ denote the three spatial dimensions, and 1 denotes the single channel of the MRI. The human genetic data $\boldsymbol{G}$ consists of $N$ positions on the human genome with frequent variants (SNPs). Each SNP $\boldsymbol{g}_i$ is represented by a four-tuple $\left(d_i, c_i, p_i^{\text{phy}}, p_i^{\text{gen}}\right)$. In the four-tuple, $d_i \in \{0, 1, 2\}$ denotes the genotype of the SNP, the number of copies of the mutant allele, $c_i \in \{1, \cdots, 22\}$ denotes the index of chromosome the SNP belongs to, $p_i^{\text{phy}} \in \mathbb{N}$ denotes the physical position in terms of base pair (bp) of the SNP in the chromosome, and $p_i^{\text{gen}} \in \mathbb{R}^+$ denotes the genetic position of the SNP in terms of centimorgan (cM). Note that the genetic data is not a sequence but an array since two neighboring SNPs $\boldsymbol{g}_i$ and $\boldsymbol{g}_{i+1}$ are not necessarily consecutive on the original genome and the physical (and genetic) distance between them $\left|p_i^{\text{phy}} - p_{i+1}^{\text{phy}}\right|$ is meaningful. We further denote arrays consisting of all genotypes, chromosomes, and positions sorted on chromosome id and physical position by $\boldsymbol{d}$, $\boldsymbol{c}$, $\boldsymbol{p}^{\text{phy}}$, and $\boldsymbol{p}^{\text{gen}}$, respectively.

## 3 What Makes Appropriate MI Estimators for GWAS?

With the goal of learning representations that capture as much information about the genetic variations as possible, our objective is to maximize the MI between the representation and genetic data with an appropriate MI estimator. One commonly used approach for estimating the mutual information between multi-dimensional variables is the Jensen-Shannon Estimator (JSE) (Nowozin et al., 2016). The JSE involves a discriminator to distinguish whether samples of the two variables belong to the same individual or are independently sampled. Specifically, under our problem setting, the JSE-based training loss is computed as

$$\begin{aligned}
\mathcal{L}_{\text{JSE}}(\boldsymbol{B}; \theta, \phi) = &-\frac{1}{|B|} \sum_{(\boldsymbol{Y}, \boldsymbol{G}) \in \boldsymbol{B}} \log\left(\mathcal{D}_\phi\big(f_\theta(\boldsymbol{Y}), \boldsymbol{G}\big)\right) \\
&-\frac{1}{|B|(|B|-1)} \sum_{(\boldsymbol{Y}, \boldsymbol{G}) \in \boldsymbol{B}} \left[\sum_{(\boldsymbol{Y}', \boldsymbol{G}') \in \boldsymbol{B} \setminus \{(\boldsymbol{Y}, \boldsymbol{G})\}} \log\left(1 - \mathcal{D}_\phi\big(f_\theta(\boldsymbol{Y}), \boldsymbol{G}'\big)\right)\right],
\end{aligned} \tag{2}$$

where $\boldsymbol{B}$ is a mini-batch of paired MRI and genetic data and $\mathcal{D}_\phi : \mathbb{R}^q \times \mathcal{G} \to (0, 1)$ is a learnable discriminator to determine whether $f_\theta(\boldsymbol{Y})$ and $\boldsymbol{G}$ are from the same individual. Together with the Noise Contrastive Estimation (InfoNCE) (Bachman et al., 2019), these learning processes are also known to be in a contrastive manner across two modalities; namely, the MRI and genetic data.

To achieve desirable performance in maximizing MI and discovering genetic associations, the discriminator should meet certain requirements. First, the learnable discriminator $\mathcal{D}_\phi$ should be able to take as inputs the genetic data $\boldsymbol{G}$ and encode all the useful information from $\boldsymbol{G}$. A well-designed genetic encoder is thus a critical component of $\mathcal{D}_\phi$. It should be able to efficiently and effectively use not only the genotypes, but also their corresponding chromosome, physical position, and genetic position information. In Section 4, we

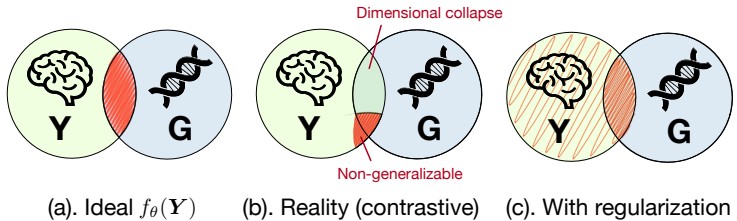

(a). Ideal $f_\theta(\boldsymbol{Y})$    (b). Reality (contrastive)    (c). With regularization

Figure 1: An illustration showing how the representation $f_\theta(\boldsymbol{Y})$ captures the mutual information between $\boldsymbol{Y}$ and $\boldsymbol{G}$ in different cases. The circles are the entropy of G and Y, respectively, and their intersection is the mutual information $\mathcal{I}(G, Y)$. Areas indicated by squiggles in red represent the information contained in $f_\theta(\boldsymbol{Y})$.

propose a novel transformer-based genetic encoder, dubbed genetic transformer, that fully utilizes all this information based on genetic intuitions.

Second, the discriminator should make predictions based on all generalizably associated patterns, rather than memorizing noise or focusing on a small portion of associated patterns that can be easily learned. However, due to the nature of contrastive learning and several differences between typical visual representation learning and GWAS representation learning, we will argue below that it is challenging to meet this requirement. The typical contrastive loss can lead to degenerated results for GWAS as shown in Figure 1.

### 3.1 Uniqueness of GWAS Representation Learning and Limitations of Contrastive Losses

To understand the limitations of typical contrastive losses in the GWAS setting, we first identify key differences between the visual representation learning problem for natural images and the GWAS representation learning on high-dimensional data. We explain how each difference can contribute to limitations or failures of typical contrastive losses in the GWAS setting, and provide empirical evidence to support our arguments.

**Difference 1: Goals of learning representations**. Typical visual representations of natural images aim to capture the key semantics or class information about major objects in images. With this goal, it is acceptable for the representation to capture only semantic or class-related information or even required that representations are invariant to elements such as context (Zhang et al., 2021; Chang et al., 2021) and transformations (Xiao et al., 2020; Foster et al., 2020). In this case, a good representation for downstream tasks does not necessarily maximize the MI during contrastive learning. In contrast, a good representation for the GWAS purpose should capture every detail or pattern in the high-dimensional MRI data that is associated with the genes, since there is no such key semantics or class information. In this case, the downstream GWAS performance is closely associated with $\mathcal{I}(f_\theta(\boldsymbol{Y}), \boldsymbol{G})$.

**Limitation 1: Dimensional collapse**. A recent study on visual representation learning (Jing et al., 2021) identifies and empirically shows that typical contrastive approaches suffer from the dimensional collapse issue, where the learned representations occupy a lower-dimensional subspace than their designated dimensions. The dimensional collapse results in high redundancy, limits the information captured by representations, and therefore leads to reduced performance in downstream tasks. Indeed, our analyses

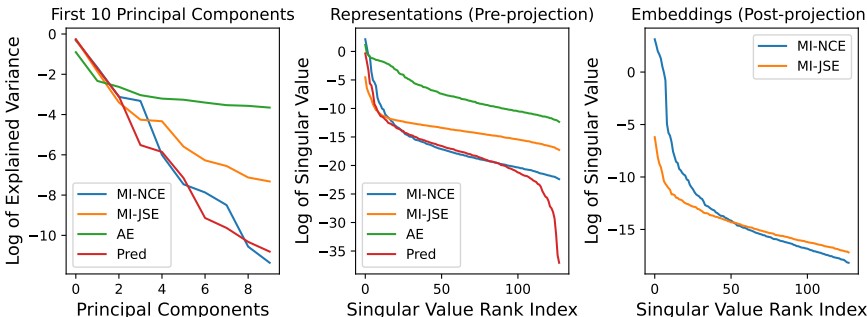

Figure 2: The logarithm of the explained variance for the first 10 principal components (**left**), the singular value spectrum of learned representations (**middle**) and embeddings after projection (**right**). Comparisons are among contrastive MI estimators with InfoNCE (MI-NCE), with JSE (MI-JSE), Autoencoder (AE), and genetic data prediction (Pred).

show that the dimensional collapse issue also arises in the cross-modal contrastive setting. We compare the singular values of representations learned by predictive methods and contrastive methods in Figure 2. Results indicate that the contrastive estimators NCE and JSE suffer from dimensional collapse with a dramatic drop in explained ratios and singular values. Even worse, the GWAS performance suffers more from the dimensional collapse issue due to its nature described in **Difference 1**, as both the information of $f_\theta(\boldsymbol{Y})$ and the mutual information $\mathcal{I}(f_\theta(\boldsymbol{Y}), \boldsymbol{G})$ is limited.

**Difference 2: Augmentation approaches and data dimensions**. Contrastive learning relies on a large number of samples to more accurately estimate and maximize the mutual information between different views or modalities. Previous studies (Chen et al., 2020; Tian et al., 2020) have shown that augmentations are crucial for contrastive learning, as they prevent representations from focusing on patterns that are irrelevant to downstream tasks and multiply the number of training samples. For higher-dimensional 3D MRI data,

more samples and diverse augmentations are necessary (Vapnik, 1999). However, the availability of medical imaging data is limited, and most augmentations used for typical visual representation learning are not applicable to medical imaging. For example, since MRIs are single-channel, color space augmentations are not possible. Augmentations based on rotation and flipping are not suitable for brain MRI data due to their asymmetric nature. Random linear transform or non-linear morph may change the shape of the elements of the image and thus are discouraged. In the case of 3D MRI, the applicable augmentations are very limited.

**Limitation 2: Non-generalizable associations**. According to Huang et al. (2021), augmentations play a critical role in the generalization capability of contrastive learning approaches. However, due to the limited number of applicable augmentation techniques and the dimensionality of 3D data, the discriminator tends to capture non-generalizable or false associations from the training samples such as memorizing the shape, the layout of the brain in the MRI, or specific noise in the data to identify individuals. In Figure 3, we evaluate the generalizability of models trained with contrastive loss by comparing the MI estimation on training and validation pairs. The remarkable discrepancy in losses between the training and validation sets suggests that the discriminator used by contrastive loss is unable to generalize to new samples, indicating that contrastive loss is a poor estimation of MI in our case.

From the perspective of mutual information, due to the dimensional collapse, a limited amount of training samples, and insufficient means of augmentations, empirical results show that the JSE is not an optimal estimator of mutual information. The true mutual information is hence not maximized by the learned representations, leading to degraded performance in GWAS. Figure 1 (a–b) illustrates the relationship among the brain MRI, genetic data, and the learned representation. In the ideal case shown in (a), the representation should perfectly cover the mutual information between $\boldsymbol{Y}$ and $\boldsymbol{G}$, so that the learning objective achieves its maximal with

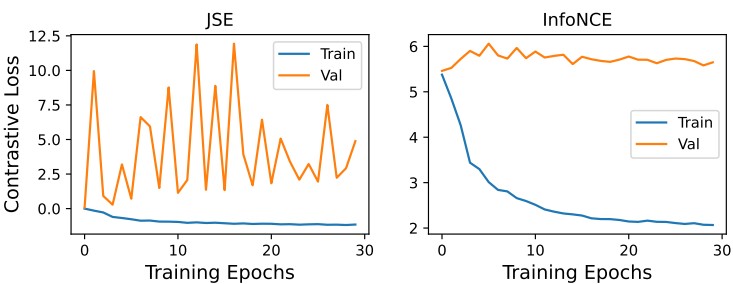

Figure 3: Generalization capability of contrastive MI estimators JSE (**left**) and InfoNCE (**right**). The learned discriminators fail to generalize to unseen pairs, leading to a large discrepancy between training and validation losses.

$$\mathcal{I}(f_\theta(\boldsymbol{Y}), \boldsymbol{G}) = \mathcal{I}(\boldsymbol{Y}, \boldsymbol{G}) \geq \mathcal{I}(f_{\theta'}(\boldsymbol{Y}), \boldsymbol{G}), \ \forall f_{\theta'}. \tag{3}$$

In practice with brain MRI data, the contrastive loss results in representations that only capture a small portion of $\mathcal{I}(\boldsymbol{Y}, \boldsymbol{G})$ due to the two limitations described above, as shown in (b).

## 3.2 MI Estimator with Regularizations

Given the issues and limitations outlined above, our goal is to improve the representation by incorporating more generalizably associated patterns in addition to those identified by the contrastive MI estimator. However, unlike in the case of natural images where many elements are known to be non-generalizable, our limited knowledge of undiscovered genetic associations makes it difficult to determine which patterns are generalizable and which are not. As a result, it is challenging to develop targeted augmentations that make the representation invariant to unwanted patterns.

To achieve our goal without requiring further knowledge, we propose to uniformly increase the total information contained in the representation by including an entropy term in the learning objective. The objective is formulated as

$$\max_\theta \left[ \hat{\mathcal{I}}(f_\theta(\boldsymbol{Y}), \boldsymbol{G}) + \lambda \mathrm{H}(f_\theta(\boldsymbol{Y})) \right], \tag{4}$$

where $\hat{\mathcal{I}}$ is the contrastive MI estimation JSE, $H$ denotes the entropy of a random variable, and $\lambda$ is a weight scalar. The entropy term encourages the representation to capture more information about $\boldsymbol{Y}$ and reduces its redundancy. A certain portion of the information can contribute to the generalizable associations, as

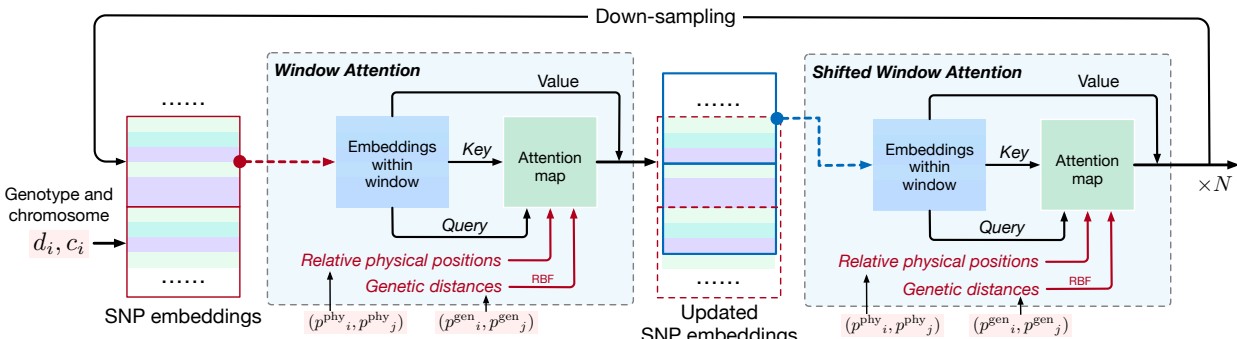

Figure 4: A swin-transformer block in the proposed genetic encoder. Red and blue boxes represent the windows and shifted windows.

illustrated in Figure 1-(c). The entropy term serves as a regularization to the estimated MI to improve its generalizability. From the optimization aspect of view, the objective is considered as adding a Lagrange multiplier to maximize the entropy $\text{H}(f_\theta(\boldsymbol{Y}))$, subject to the constraint that the estimated mutual information $\hat{\mathcal{I}}(f_\theta(\boldsymbol{Y}), \boldsymbol{G})$ achieves its maximum. When multiple patterns can be used to identify individuals, the entropy term encourages the model to capture as many of them as possible, instead of capturing the easiest ones.

There are various methods to estimate and optimize the entropy, such as minimizing the off-diagonal values in the covariance matrix of the representation (Zbontar et al., 2021). However, these estimations require a large mini-batch size, which is not suitable in our case due to memory constraints caused by the 3D data and MRI encoder. As an alternative, we use the reconstruction of MRI data as a proxy to maximize the entropy. The loss is then computed as

$$\mathcal{L}(\boldsymbol{B}; \theta, \phi, \psi) = \mathcal{L}_{\text{JSE}}(\boldsymbol{B}; \theta, \phi) + \frac{\lambda}{|B|} \sum_{(\boldsymbol{Y}, \boldsymbol{G}) \in \boldsymbol{B}} \|\boldsymbol{Y} - h_\psi(f_\theta(\boldsymbol{Y}))\|^2, \tag{5}$$

where $h_\psi$ is a deterministic decoding head used to reconstruct the MRI from the representation $f_\theta(\boldsymbol{Y})$. Compared to other proxies discussed by Zbontar et al. (2021), the reconstruction term is less sensitive to small mini-batch sizes. To justify the reconstruction term, we have

$$H(f_\theta(\boldsymbol{Y})) = \mathcal{I}(f_\theta(\boldsymbol{Y}), \boldsymbol{Y}) + \underbrace{H(f_\theta(\boldsymbol{Y})|\boldsymbol{Y})}_{0}, \tag{6}$$

where the conditional entropy $H(f_\theta(\boldsymbol{Y})|\boldsymbol{Y})$ is reduced to zero when the encoder $f$ is deterministic. Intuitively, given the input Y, the learned embedding f(Y) will not carry additional information about Y. We then consider the maximization of the mutual information $\mathcal{I}(f_\theta(\boldsymbol{Y}), \boldsymbol{Y})$. According to Wang and Isola (2020), the objective consists of two components: 1) the alignment of $f_\theta(\boldsymbol{Y})$ to $\boldsymbol{Y}$, which is achieved by maximizing the likelihood that $f_\theta(\boldsymbol{Y})$ and $\boldsymbol{Y}$ in a positive pair belong to the same individual, and 2) the uniformity of $f_\theta(\boldsymbol{Y})$, which is achieved by maximizing the likelihood that $f_\theta(\boldsymbol{Y})$ and any other $\boldsymbol{Y}'$ are independent. To reduce the batch-size sensitivity and optimize memory cost for extremely high-dimensional MRI data, we avoid introducing additional encoders and discriminators used by conventional MI estimators. Instead, we use the reconstruction term to maximize the log-likelihood that $f_\theta(\boldsymbol{Y})$ and $\boldsymbol{Y}$ in a positive pair belong to the same individual, corresponding to the first component. The uniformity of $f_\theta(\boldsymbol{Y})$ is addressed in $I(f_\theta(\boldsymbol{Y}), \boldsymbol{G})$ by maximizing the likelihood that $f_\theta(\boldsymbol{Y})$ and any other $\boldsymbol{G}'$ are independent, intuitively ensuring that the embeddings of any two different individuals are distinguishable from each other. From the perspective of Wang and Isola (2020), the two terms in $\mathcal{L}_{\text{JSE}}(\boldsymbol{B}; \theta, \phi)$ and the reconstruction term aim at optimizing three properties of $f_\theta(\boldsymbol{Y})$; namely, the alignment to $\boldsymbol{G}$, the uniformity, and the alignment to $\boldsymbol{Y}$.

## 4 Genetics-Informed Transformer

A typical current genotyping microarray of the human genome includes more than 650k SNPs, with the physical spacing between any two consecutive SNPs being inconsistent. Genetic encoders developed in existing

studies (van Hilten et al., 2021; Taleb et al., 2022) based on convolutional neural networks (CNNs) and multi-layer perceptrons (MLPs) are incapable of handling the unstructured genetic data with extremely large sizes. To address this, we develop an effective genetics-informed transformer to encode genetic data in accordance with the following objectives:

1. Significant optimized computational cost, in recognition of certain biological assumptions.
2. Information aggregation among SNPs from arbitrary positions in the genome, considering multiple genetics dependency measurements.
3. Flexibility to accept any segments or subsets of the genetic data as input, thereby facilitating cropping or downsampling-based augmentations on the genetic data.

An overview of the proposed transformer block is shown in Fig. 4, where attention operators with shifting windows (Liu et al., 2021) are used to enable efficient computation, and the aggregation is specialized with both physical and genetic distances of SNPs. In the transformer, two blocks are connected by down-sampling with attention-based pooling operators, and the initial SNP embeddings are computed based on both the genotypes and the chromosome each SNP belongs. Finally, an attention-based readout is used to compute the global representation.

**Window attention in swin-transformer.** The 1D swin-transformer performs self-attention operations within each window split from the entire genetic array to enable efficient computing. It contains two components; those are, window attention and shifted window attention, as shown in Fig. 4. Given input SNP embeddings $\boldsymbol{H} \in \mathbb{R}^{L \times q}$ where $L$ denotes the number of SNPs and $q$ denotes the embedding dimension, window attention first splits $\boldsymbol{H}$ into a set of windows $\{\boldsymbol{H}_i \in \mathbb{R}^{w \times q}\}_{i=1,\cdots,\lfloor L/w \rfloor}$ where each window has a size $w$. A self-attention block is then applied to each $\boldsymbol{H}_i$ to update the SNP embeddings by aggregating information within that window. The updated windows are then merged back following the order when splitting $\boldsymbol{H}$, forming $\boldsymbol{H}'$ as the final output of the window attention component. In the following shifted window attention component, $\boldsymbol{H}'$ is first shifted by a length of $\lfloor w/2 \rfloor$. Then similar splitting and self-attention are performed as in window attention to update SNP embeddings from each individual window. Finally, the updated windows are merged back, and the merged sequence is also shifted back by a length of $\lfloor w/2 \rfloor$. There exists a biological assumption that strong and informative dependencies between SNPs exist only when they are within a certain distance. Hence, compared with performing global attention on all 650k marker positions, performing attention within windows in our proposed methods aggregates similar information, but largely reduces the computing cost.

**Aggregation based on multiple dependencies.** Multiple attention heads are computed to capture various types of dependencies among markers. Specifically, the computing of attention scores captures dependencies from three perspectives; those are, the SNP embeddings reflecting potential co-mutation, the encodings of the physical positions of SNPs on a chromosome indicating local dependencies among SNPs, and the encodings of genetic positions of SNPs measuring genetic linkages. Formally, the attention score $\alpha_{i,j}^k$ between the $i$-th and $j$-th SNPs for the $k$-th attention head is computed as

$$\alpha_{i,j}^k = f_\alpha^k(\boldsymbol{g}_i, \boldsymbol{g}_j) = f_{\alpha_g}^k(\boldsymbol{h}_i, \boldsymbol{h}_j) + f_{\alpha_p}^k(p_i^{\text{phy}}, p_j^{\text{phy}}) + f_{\text{RBF}}^k(p_i^{\text{gen}} - p_j^{\text{gen}}), \tag{7}$$

where $\boldsymbol{h}_i = \boldsymbol{H}[i] \in \mathbb{R}^q$ denotes the SNP embedding. We compute the three individual components of attention scores that can capture different genetic dependencies as

$$f_h^k(\boldsymbol{h}_i, \boldsymbol{h}_j) = \left(\boldsymbol{h}_i^T \boldsymbol{W}_h^{k,l}\right)\left(\boldsymbol{h}_j^T \boldsymbol{W}_h^{k,r}\right)^T, f_{\text{pe}}^k(p_i^{\text{phy}}, p_j^{\text{phy}}) = \left[\boldsymbol{e}_{\text{p}}(p_i^{\text{phy}})^T \boldsymbol{W}_{\text{pe}}^{k,l}\right]\left[\boldsymbol{e}_{\text{p}}(p_i^{\text{phy}})^T \boldsymbol{W}_{\text{pe}}^{k,r}\right]^T,$$

$$f_{\text{RBF}}^k(p_i^{\text{gen}} - p_j^{\text{gen}}) = \left[\boldsymbol{r}\left(\left|p_i^{\text{gen}} - p_j^{\text{gen}}\right|\right)\right]^T \left[\mathbb{1}_{(p_i^{\text{gen}} - p_j^{\text{gen}}) \geq 0}^T \boldsymbol{W}_{\text{rbf}}^{k,+} + \mathbb{1}_{(p_i^{\text{gen}} - p_j^{\text{gen}}) < 0}^T \boldsymbol{W}_{\text{rbf}}^{k,-}\right], \tag{8}$$

where $\boldsymbol{e}_p(\cdot)$ denotes the position encoding (Devlin et al., 2019), $\boldsymbol{W}_g^{k,l}, \boldsymbol{W}_g^{k,r}, \boldsymbol{W}_{\text{pe}}^{k,l}, \boldsymbol{W}_{\text{pe}}^{k,r}$, and $\boldsymbol{W}_{\text{rbf}}^{k,+}, \boldsymbol{W}_{\text{rbf}}^{k,-}$ are trainable projections. $\mathbb{1}_{condition}^T$ is an indicator vector where all elements are 1s if the condition holds, and are 0s otherwise. The function $\boldsymbol{r}$ denotes a distance expansion with radial basis functions (RBF) (Schütt et al., 2017). Denoting $s := \left|p_i^{\text{gen}} - p_j^{\text{gen}}\right|$, the term $\boldsymbol{r}\left(\left|p_i^{\text{gen}} - p_j^{\text{gen}}\right|\right)$ in the above equation is computed as

$$\boldsymbol{r}(s) = \left[\exp\left\{(s-t)^2/\sigma^2\right\}\right]_{t \in \{t_0, \cdots, t_c\}} \in \mathbb{R}^c, \tag{9}$$

where $\{t_0, \cdots, t_c\}$ is a set of non-negative real numbers ranging from 0 to a preset threshold. The asymmetric projections in all $f_h, f_{pe}$ and $f_{RBF}$ functions indicate that $\alpha_{i,j}$ does not necessarily equal to $\alpha_{j,i}$, leading to more expressive models to capture dependencies. In addition, the computing of each attention head is based on a combination of three types of dependencies, which enables information aggregation among different SNPs based on genetic dependencies. By doing this, the complicated genetic dependencies of the input genome data can be captured. Additional details about the transformer are provided in Appendix B.

## 5 Related Work

**Dimension reduction for GWAS**. When doing genome-wide association studies, people oftentimes find themselves dealing with high-dimensional quantitative traits. In order to reduce computational cost and redundancy, and in the hope of finding meaningful underlying patterns, many works perform dimension reduction of the high dimensional traits before doing GWAS, including principal component analysis (Zhao et al., 2021; Yano et al., 2019; Ma et al., 2021), independent component analysis (Elliott et al., 2018; Pearlson et al., 2015) and non-negative matrix factorization (Wen et al., 2022). These approaches are effective in capturing linear dependencies but are less capable of identifying complicated traits from imaging data.

**Deep learning-based approaches**. Recently, several works used unsupervised learning to characterize high-dimensional medical data. iGWAS (Xie et al., 2022) applied contrastive learning between multiple images of the same person to reveal potential genetic signals, ContIG (Taleb et al., 2022) applied contrastive learning between medical imaging data and genetic data to learn the feature representation. DeepEndo autoencoder (Patel et al., 2022) used a convolutional autoencoder to reduce the dimensionality of the imaging data and found genetic associations of these extracted phenotypes. TransferGWAS (Kirchler et al., 2022) used both supervised task and reconstruction task to learn the feature representation. Specifically, ContIG (Taleb et al., 2022) is the first to use contrastive learning between images and genetics on the GWAS problem. However, there are distinguishable differences between the work and ours. First, ContIG aims to learn general representation for multiple downstream tasks, such as classifications of the risk of several diseases. With this goal, ContIG treats the problem as a typical visual representation learning task. On the contrary, our study focuses on the representation learning specifically for GWAS. Second, our approaches are built upon the grounding of mutual information maximization, whereas ContIG is grounded by contrastive learning for unlabelled data. Third, our work focuses on a more challenging setting with 3D MRI data, where typical contrastive approaches may fail.

**Mutual Information Maximization**. Previous research has employed mutual information maximization as a pretext task for representation learning on various data types, including images (Hjelm et al., 2019), videos (Hjelm and Bachman, 2020), and graphs (Veličković et al., 2019; Stärk et al., 2022). However, these studies primarily focus on classification or regression as downstream tasks. Our work presents unique challenges and goals of mutual information maximization under the GWAS setting and we are the first to examine GWAS from a mutual information perspective.

## 6 Experiments

We use the brain imaging dataset from UK Biobank (Sudlow et al., 2015) in this study, it is currently the largest public brain imaging dataset. Specifically, we use T1-weighted MRI imaging data accessed on October 15, 2021. We register and pre-process the MRI data into the shape of $182 \times 218 \times 182$. For representation learning, we split the MRI-genetic data pairs into 4,597 training and 1,533 validation pairs based on ethnicity. A detailed description of the processing and split is provided in Appendix A.

**GWAS Evaluation Metrics** We involve three metrics to evaluate the representations learned by different models; namely, the **number of loci** discovered by GWAS, the **estimated mutual information**, and the **heritability** of the representations. All three metrics are computed on a testing dataset that is unseen during the representation learning process and measures the quality of representation for GWAS purposes. To enable efficient evaluation, we obtain the first 10 principal components of representations and compute all metrics on the 10-dimensional vectors. We provide detailed descriptions of each evaluation metric in Appendix A.

Table 1: Comparisons of quantitative evaluation results on the test set. # Loci are counted under Bonferroni corrected p-value threshold (cutoff) of 5e-9. "Unique" refers to the number of loci discovered by a method that is NOT discovered by any methods in other groups.

|  | Methods | #Independent Loci | | Estimated MI↑ | Heritability $h^2$ ↑ |
|  |  | All↑ | Unique↑ |  |  |
|---|---|---|---|---|---|
|  | Random Init | 14 | 1 | $1.2178 \pm 0.0128$ | $0.0756 \pm 0.0656$ |
| *Predictive* | Autoencoder (Patel et al., 2022) | 26 | 1 | $1.3124 \pm 0.0062$ | $0.3121 \pm 0.0769$ |
|  | Autoencoder-attention | 23 | 4 | $1.3138 \pm 0.0061$ | $0.2984 \pm 0.0773$ |
|  | Gen Prediction | 10 | 0 | $1.2482 \pm 0.0116$ | $0.0918 \pm 0.1110$ |
| *Contrastive* | Barlow Twins (Zbontar et al., 2021) | 11 | 1 | $1.3008 \pm 0.0072$ | $0.0814 \pm 0.0636$ |
|  | SimCLR (Chen et al., 2020) | 15 | 1 | $1.2355 \pm 0.0131$ | $0.1448 \pm 0.1128$ |
|  | SimCLR-JSE | 17 | 7 | $1.3062 \pm 0.0068$ | $0.1604 \pm 0.1151$ |
| *Trans-Modal Contrastive* | InfoNCE (ContIG, Taleb et al. (2022)) | 11 | 0 | $1.2268 \pm 0.0142$ | $0.1334 \pm 0.0588$ |
|  | Decorrelated InfoNCE | 13 | 3 | $1.2274 \pm 0.0128$ | $0.0527 \pm 0.0349$ |
|  | GIM (Ours) | **40** | **15** | $\mathbf{1.3782 \pm 0.0010}$ | $\mathbf{0.3807 \pm 0.0517}$ |

## 6.1 Quantitative Results and Analysis

We compare our approach with multiple baseline approaches in four groups; namely, MRI encoders that are randomly initialized, trained by predictive approaches, contrastive approaches with MRI data only, and trans-modal contrastive approaches that involve genetic data. The baselines include existing or straightforward training schemes, namely, Autoencoder (Kirchler et al., 2022; Patel et al., 2022), `Gen Prediction` that uses genetic data prediction as a pretext task, Barlow-Twins (Zbontar et al., 2021), SimCLR (Chen et al., 2020), and ContIG (Taleb et al., 2022). We additionally include their variants `Autoencoder-attention` that uses the same MRI encoder as ours, `SimCLR-JSE`, where the contrastive objective in SimCLR is replaced by JSE, and `Decorrelated InfoNCE`, where a decorrelation term is added to the contrastive loss. We describe the implementation details of GIM and baselines in Appendix B.

Comparisons among representations learned by different methods in terms of the three metrics are shown in Table 1. The results indicate that the proposed learning framework with a regularized MI estimator and genetic transformer significantly improves the quality of learned representation in terms of the number of discovered loci and the heritability. The improved MI of our methods on test pairs also suggests a stronger generalization capability. Additionally, we have the following observations.

**The level of mutual information on test pairs agrees with # loci and heritability**. The results suggest that higher mutual information on the test set implies a higher heritability and more loci discovered. It justifies our formulation of learning representation for GWAS as the problem of maximizing mutual information.

**Typical trans-modal contrastive approaches fail for MRI data**. Trans-modal contrastive learning with typical contrastive loss performs fairly well on 2D retina imaging (Taleb et al., 2022) but suffers more from the performance reduction on the higher-dimensional 3D data. In the 3D MRI case, we found that the simplest Autoencoder approach performs even better than contrastive and typical trans-modal contrastive approaches.

**GIM has consistent performance**. To further examine the robustness of GIM, we obtain additional results by training 3 additional GIM models independently at different randomizations. They resulted in 43, 42, and 46 discovered loci, respectively, which indicates a consistent and significant improvement compared to baseline approaches.

## 6.2 Ablations and Additional Results

We perform additional quantitative studies to demonstrate the effectiveness of GIM. Visualizations for representation uniformity and Manhattan plot for GWAS are provided in Appendices C, D, and E.

**Effectiveness of individual proposed components**. To show the effectiveness and necessity of both the proposed learning objective and the genetic transformer, we track the change in the number of all loci, unique

Table 2: Change in the number of loci and heritability when incrementally adding components to the models.

| Methods | # Loci | Change | # Unique Loci | Change | $h^2$ | Significant Gene |
|---|---|---|---|---|---|---|
| Base-Contrastive | 11 | - | 0 | - | $0.1334 \pm 0.0588$ | - |
| + Regularization | 29 | +18 | 6 | +6 | $0.3390 \pm 0.0627$ | *CENPW* |
| + Genetic Transformer | 32 | +3 | 10 | +4 | $0.3773 \pm 0.0384$ | *WNT16* |
| + Random Cropping | 36 | +4 | 13 | +3 | $0.3723 \pm 0.0558$ | *ITPR3* |
| + Co-training | 40 | +4 | 15 | +2 | $0.3807 \pm 0.0517$ | *MSRB3* |

loci, the heritability score, and newly discovered genes with the highest significance when incrementally adding each component. Table 2 shows the results of adding the regularization to the objective, replacing the MLP encoder with the genetics-informed transformer, and performing random cropping on the genetic data. The results suggest that adding each component generally increases the useful information carried by the representations, leading to more loci discovered. We discuss the new gene discovered by each component and the change of $h^2$ in Appendix D.

**GWAS representation for T2-weighted MRI**. We additionally apply GIM to a second modality, namely the T2-weighted MRI. Similarly, we compute GWAS on the first 10 principle components of the learned representation on the test set. In contrast to the results for T1, we observe that learning informative representations is less challenging for contrastive methods in the T2 case. In addition, contrastive methods equipped with NCE generally perform better than their JSE counterparts. This result is consistent with those presented in Taleb et al. (2022). Nevertheless, results in Table 3 show a consistent out-performance of GIM over baselines, indicating generalizable effectiveness.

Table 3: # Loci for T2-weighted MRI at cutoffs 5e-9 and 1e-8.

| Methods | 5e-9 | 1e-8 |
|---|---|---|
| Autoencoder | 29 | 48 |
| Barlow-Twin | 7 | 14 |
| SimCLR | 21 | 28 |
| SimCLR-JSE | 15 | 20 |
| ContIG | 22 | 29 |
| GIM (Ours) | **38** | **62** |

**Downstream classification performance**. Although this is not our primary goal, we perform additional evaluation with three downstream binary classification tasks, namely stroke, cancer, and biological sex, from UK Biobank. We follow Taleb et al. (2022) to perform linear evaluation protocol and compute their ROC-AUC. Specifically, we obtain MRI embedding from individual MRI encoders. We then freeze all embeddings and perform linear classification upon the embeddings for the three tasks. Since ROC-AUC is computed on the entire validation set, we repeat the classification on 5 random training-validation splits and show their average ROC-AUC score. We include ablations for GIM with or without the presence of regularized loss (Reg.), Genetics-Informed Transformer (GT), and random cropping (Cropping) with different probabilities. The results shown in Table 4 indicate that GIM achieves better performance compared to its contrastive counterparts. For cancer and biological sex classification, we identify continuous performance gain when adding individual components, whereas, for stroke, the regularized loss and more intensive cropping bring the most improvement. For Autoencoder, we observe significantly better performance on biological sex classification. The underperformance of cross-modal approaches is due to the exclusion of the sex chromosome in the given SNP data, and thus the MI maximization discourages the model from capturing biological sex-related information from MRI data.

Table 4: ROC-AUC of downstream classification.

| Methods | Reg. | GT | Cropping | Stroke | Cancer | Sex |
|---|---|---|---|---|---|---|
| Autoencoder | | | | $\underline{0.6748 \pm 0.0026}$ | $0.5828 \pm 0.0005$ | $\mathbf{0.9039 \pm 0.0003}$ |
| SimCLR | | | | $0.6329 \pm 0.0015$ | $0.5797 \pm 0.0005$ | $0.7739 \pm 0.0217$ |
| ContIG | | | | $0.6034 \pm 0.0001$ | $0.5385 \pm 0.0038$ | $0.5927 \pm 0.0002$ |
| GIM | ✓ | | | $0.6563 \pm 0.0069$ | $0.5738 \pm 0.0006$ | $0.7515 \pm 0.0057$ |
| GIM | ✓ | ✓ | | $0.6510 \pm 0.0002$ | $0.5789 \pm 0.0007$ | $0.8601 \pm 0.0010$ |
| GIM | ✓ | ✓ | $p = 0.2$ | $0.6387 \pm 0.0180$ | $\underline{0.5836 \pm 0.0013}$ | $\underline{0.8650 \pm 0.0016}$ |
| GIM | ✓ | ✓ | $p = 0.5$ | $\mathbf{0.6772 \pm 0.0250}$ | $\mathbf{0.5898 \pm 0.0009}$ | $0.8460 \pm 0.0161$ |

## 7    Limitation and Future Directions

Current GWAS relies on statistical testing based on linear models to identify significant loci and compute the heritability scores. However, these methods can be limited in their ability to capture non-linear and complex associations, leading to missed loci and inconsistencies between mutual information and other metrics. To address these limitations, new approaches to discovering non-linear associations are needed.

Regarding mutual information optimization, our current mutual information optimization solution aims to improve representations by capturing more generalizable associations, but it is still limited in reducing information on non-generalizable ones due to a lack of knowledge about non-generalizable patterns. With specific domain knowledge, it is possible to further improve these representations.

The interpretability is a further step towards understanding the association between gene and brain phenotypes and is another important future direction of this work. In fact, the learning of informative representations with "good" enough encoders is a crucial prerequisite of interpretability. Once such an encoder model becomes available, we are able to adapt existing visual interpretation or voxel-level explanation approaches to identify significant regions in the brain that are associated with certain genes. The interpretability on the genetic encoder side is equally important. While capturing some genetic information, it lacks the resolution to trace back the contributions from individual SNP markers. Future work is needed to develop a more efficient and high-resolution genetic encoder.

## 8    Conclusions

In this work, we have investigated the differences and limitations of GWAS representation learning compared to typical visual representation learning and have presented Genetic InfoMax, a GWAS representation learning framework. We have established standardized evaluation protocols to benchmark existing and our approaches. Our experiments demonstrate a significant boost in GWAS performance by GIM.

## Acknowledgement

This work was supported by the National Institutes of Health grant U01AG070112.

## Broader Impact Statement

The use of individualized genetic data and imaging data in medical research raises ethical concerns related to privacy, informed consent, and potential discrimination. The data used in this study were from the UK Biobank under the approved *Project 24247*. The Committee for the Protection of Human Subjects of The University of Texas Health Science Center at Houston gave ethical approval for this work under *HSC-SBMI-20-1323*. We followed the best practice and confirmed that all necessary patient/participant consent has been obtained and the appropriate institutional forms have been archived, and that any patient/participant/sample identifiers included were not known to anyone (e.g., hospital staff, patients, or participants themselves) outside the research group so cannot be used to identify individuals. Participants enrolled in the UK Biobank have provided broad consent, expressly permitting the use of their data for biomedical research purposes that encompass the objectives of this study, ensuring compliance with ethical standards and the intended broad impact of our research.

The work focuses on the development of new methods for uncovering genetic factors contributing to brain morphology as measured by brain MRI images. Our model training sets include people from diverse ancestry backgrounds. The GWAS cohorts include primarily self-reported British white individuals. Therefore, the generalizability of the genetic findings to other populations and people of different ancestries is not tested. Future studies of more diverse samples are warranted. Since the brain morphological features derived are not of immediate medical use, the risk of perpetuating existing healthcare disparities is indirect.

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

# A  Data Processing and Split

All MRIs were linearly registered (affine registration with 12 DOF) to standard MNI152 space using the UKBiobank-provided transformation matrix with FSL FLIRT (Jenkinson et al., 2002) and all the outputs are of shape 182×218×182. A large portion of the UKBiobank population are white British. In order to maximize the power of genetic discovery and avoid the complication of population stratification, the genetic association study was only done on the white British (UKBiobank data field 21000 and 22006) cohort. So we selected 6,130 images from subjects of mixed ethnicities (all non white British samples plus a small number of random white British samples) not overlapped with the samples for the genetic discovery to do the training and validation, among which 4,597 was randomly selected for training and 1,533 for validation. We used two quality metrics "inverted contrast-to-noise ratio" (UKBiobank data field 25735) and "Discrepancy between T2 FLAIR brain image and T1 brain image" (UKBiobank data field 25736) to ensure the quality of the training data.

Details about the evaluation metrics are provided below.

**Number of Loci** We perform genome-wide scans over 658,720 directly genotyped SNPs [1] and on 28,489 white British participants unseen during training. We use BOLT-LMM (Version 2.3.4) (Loh et al., 2015) for running GWAS. Age, gender, and the first 10 ancestral principal components are used as covariates. We use Bonferroni corrected p-value threshold of 5e−9 and a minor allele frequency threshold of 1% to get the significant SNPs and filter out the rare variants. We then cluster the significant SNPs into loci using a 250 kb window, which is approximately 0.25 cM (Elliott et al., 2018). The number of loci indicates the amount of genetic contribution to the learned features.

**Heritability** measures the proportion of variation of the feature explained by the genetic factors. It provides insight into the genetic basis of a feature. A higher heritability indicates that the representation is better associated with the genetic data. The heritability is computed using LDSC v1.0.1 (Bulik-Sullivan et al., 2015).

**Mutual Information** We estimate the mutual information between MRI representations and genetic data on the test set to explicitly demonstrate that the proposed objective adds to the generalizability of captured associations to unseen pairs. We train individual JSE-based mutual information estimators with the same architecture described in Appendix B for different methods. We train the MI estimator until the contrastive loss converges and take the opposite of the converged value as the MI estimation for each model.

# B  Implementation Details

**MRI encoders** The MRI encoder is constructed as a 3D convolutional network consisting of three residual blocks (Wang et al., 2021) connected by two downsampling operators with stride convolutions. The numbers of channel maps are 32, 64, and 128, respectively for the three blocks. The final representations are 128-dimensional computed by a dense layer upon flattened feature maps. When computing the reconstruction loss, we include additional 128-dimensional vectors computed from a multi-head attentive readout from feature maps, and the reconstruction is performed on the 256-dimensional representation after concatenation. However, dimensions from attention are not used in GWAS computation. This is to further prevent the encoder from learning too detailed patterns, possibly noise, that are non-generalizable to the test set.

**Genetic encoders** Our genetic encoder consists of three 1D swin-transformer blocks connected by two down-sampling operators with a down-sampling rate of 10. The positional encoding for SNP physical positions is 128. The embedding dimensions are 32, 64, and 128 for the three blocks, respectively. The window size to perform attention is 10 and the number of heads is 4 for all self-attention operators. The downsampling operator computes the attention with learnable queries within each window, where the window size is equal to the downsampling size. The global pooling operators compute attention with learnable queries among all positions at multiple scales and resolutions.

---

[1]Applied Biosystems UK BiLEVE Axiom Array, UKBiobank data field 22438

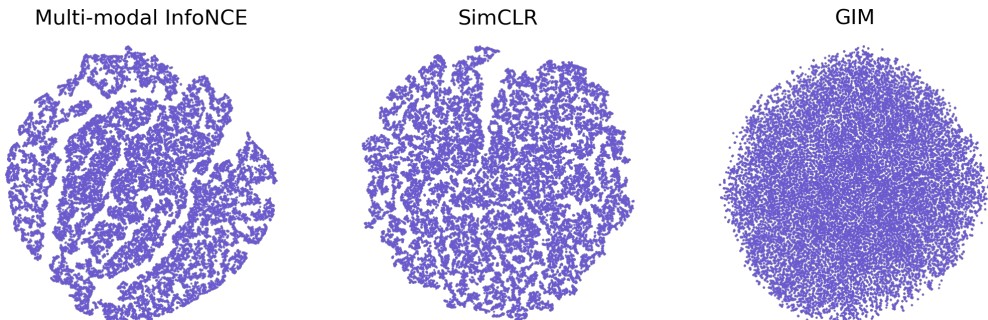

Figure 5: Visualization of learned representations with t-SNE. Forming clusters is not desired in the GWAS setting. Instead, a higher uniformity better utilizes the space capacity and could lead to better GWAS discovery.

**Training** The models are implemented with PyTorch (Paszke et al., 2019) and are trained on a single Nvidia A100 GPU. The training is performed with the Adam optimizer (Kingma and Ba, 2014), cosine annealing scheduler (Loshchilov and Hutter, 2016) with a starting learning rate of 0.001 and the mini-batch size of 12. We simply set $\lambda$ in the objective to 1 and do not exhaustively tune it. During training, we randomly crop the 3D MRI into smaller patches of size $[160, 160, 160]$. We first train the models with the genetic encoder frozen for 200 epochs, then include the augmentation on genetic data and continue training for an additional 100 epochs, and finally co-train both MRI and genetic encoders and projection heads for 50 epochs with augmentation on genetic data. To perform augmentation, the genetic data has a probability of 0.2 to be randomly cropped and a probability of 0.8 to be evenly down-sampled into a length of 65,000.

**Baseline approaches** For Gen Prediction, we apply a linear layer to the output representation as a prediction head $h^g : \mathbb{R}^q \to |G|$ to predict the class of each genotype in the genetic data and optimize the following loss.

$$\mathcal{L}_{\text{GenPred}} = \text{Cross-Entropy}\Big(h^g\big(f_\theta(\boldsymbol{Y})\big), \boldsymbol{d}\Big).$$

For baseline trans-modal contrastive methods, we follow the architecture, training loss, and training settings in Taleb et al. (2022). For the MRI augmentations, we perform the random flipping and rotation on the x-z plane, along with the random 3D patching. However, we found the flipping and rotation do not help the GWAS performance in the 3D MRI case. For correlated InfoNCE, we compute the covariance matrix of learned MRI representations and minimize its difference with the identity matrix,

$$\mathcal{L}_{\text{decor}} = ||\hat{\boldsymbol{z}}^T \hat{\boldsymbol{z}} - \boldsymbol{I}||^2,$$

where $\hat{\boldsymbol{z}}$ is the normalized MRI representations in the mini-batch. Since the mini-batch size is small due to memory constrain, the covariance estimation can be less accurate, still leading to reduced performance.

## C    Distribution of learned representations

We visualize the distribution of representations learned by trans-modal InfoNCE, SimCLR, and GIM, respectively, with t-SNE in Figure 5. Compared to baseline approaches, GIM learns representations that are more uniformly distributed in the space. According to the discussion on the difference between learning goals, the goal of our representation learning is not to form clusters for downstream classification purposes but to uniformly encode as much information about the genetic data as possible (Wang and Isola, 2020). Under this setting, clusters of representations are not desired and may harm the GWAS performance due to reduced capacity for other characteristics.

## D    Gene mapping and query for previous known associations

We mapped significant SNPs to genes using Plink v1.9 (Purcell et al., 2007), and we presented the genes that are associated with the most significant new SNP of each model in the ablation study in Table 2. *CENPW*

is known to associated with neurogenesis (Aygün et al., 2021) and cortical morphology (Sun et al., 2022), *WNT16* with skull and brain shape (Gori et al., 2015; Medina-Gomez et al., 2021), *ITPR3* with neuropathy (Rönkkö et al., 2020) and many psychiatric disorders (Cabana-Domínguez et al., 2022) and *MSRB3* with Alzheimer's (Adams et al., 2017). For each unique locus in the best-performing model, we did a range query using the GWAS Catalog (MacArthur et al., 2017) API for the previously identified brain-related associations and the result is shown in Figure 6. We also queried each locus in the result of the Big40 study (Smith et al., 2021; Elliott et al., 2018), which uses thousands of conventional image-derived phenotypes to do GWAS and we found a locus not presented in the Big40 study in Chromosome 2, base pair 218466221 to 218604356 (in hg19 coordinate). This locus is mapped to *DIRC3*, which has been shown to be associated with Alzheimer's disease (Naj et al., 2022; Wang and Li, 2021). This showcases the potential of our method in capturing features missed by the traditional expert-defined pipelines.

For the heritability, we see that the score generally increases when adding each component to the model. The only exception is the random cropping without co-training of MRI and genetic encoder. This is due to that without co-training, the genetic encoder has limited capability of adapting to the cropped genetic data from different locations. However, the cropping augmentation can still help the learning of a better MRI encoder that identifies more loci, and co-training leads to further improvement.

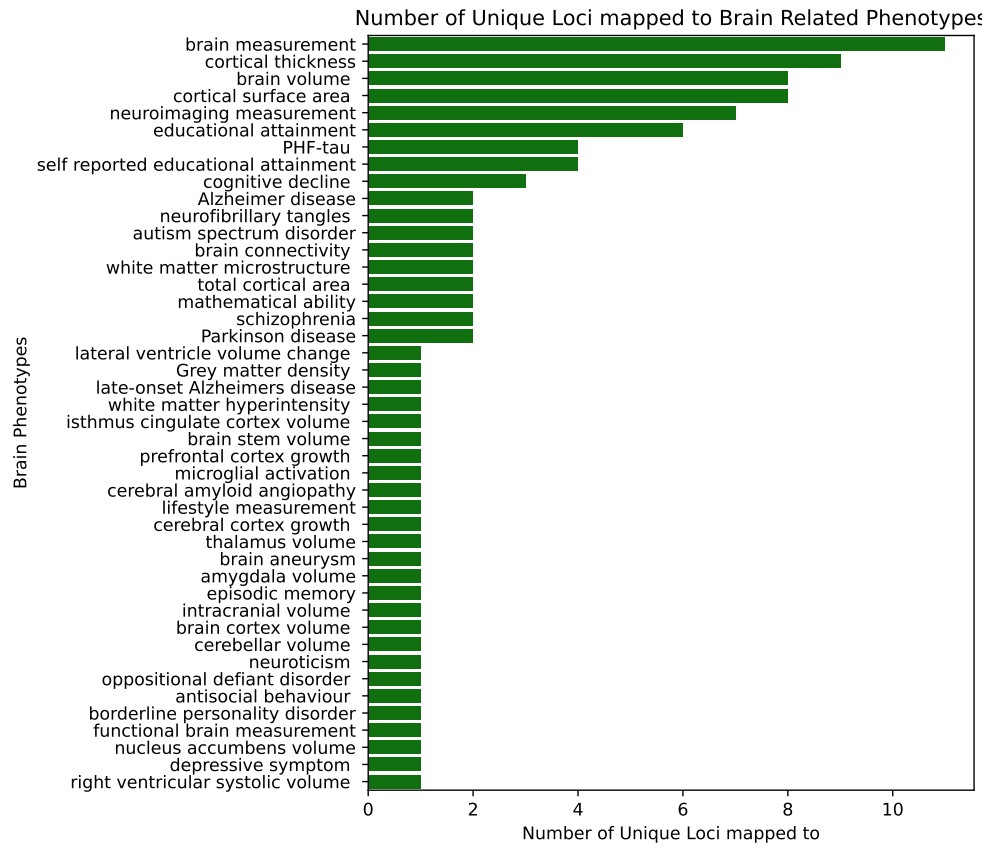

Figure 6: The number of unique loci found by the best model associated with a subset of brain-related traits in the GWAS Catalog.

## E   Manhattan Plot of GIM results

The Manhattan Plot for loci discovered by GIM is shown in Figure 7.

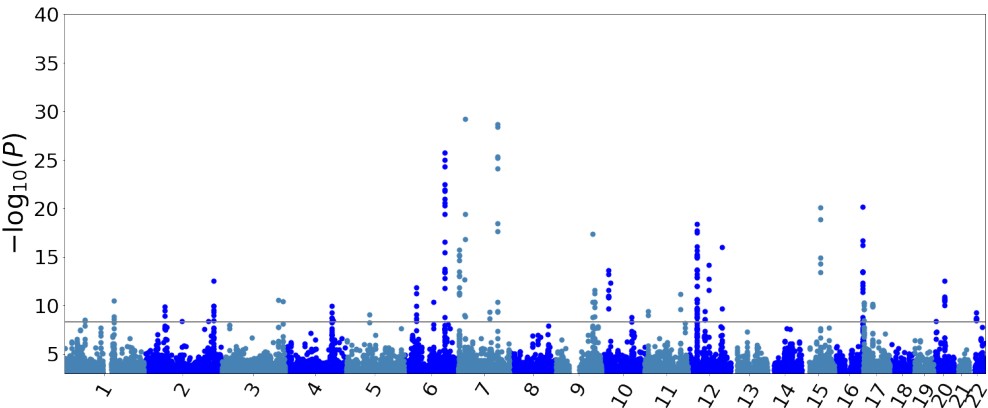

Figure 7: Aggregated Manhattan plot of the 10 PCs from the best model, the grey line represents the Bonferroni corrected p-value threshold of 5e−9.

