# OpenReview forum: "Genetic InfoMax: Exploring Mutual Information Maximization in High-Dimensional Imaging Genetics Studies"
_TMLR — Accepted by TMLR_

### Review · Reviewer_3bDZ · 2023-11-30

**Summary Of Contributions:**

The authors present a method for extracting features from 3D MRI scans that have strong associations with genetic data (SNPs).
The method relies on maximizing a combination of (i) the Mutual Information between the image representations and the genetic data as estimated by the Jensen-Shannon Estimator (JSE) and (ii) a term measuring how well the original image can be reconstructed from its lower-dimensional representation (low mean squared error).
The JSE consists in evaluating how well a model can discriminate pairs of (image features, genetic features)  that belong vs do not belong to the same individual.

The authors conduct experiments on the UKBiobank.
Image representations are learned using 4.5K individuals for training and 1.5K individuals for validation.
Then, representations are extracted for 28K unseen individuals and used to perform GWAS, looking for associations between genetic data and loadings on the 10 first principal components of the image representations.
They report the number of loci detected by the GWAS, the heritability of the PC loadings (proportion of variation explained by genetic features), and the estimated MI (with the same JSE approach) between genetic and imaging features.

The authors find that for those metrics (number of loci, estimated MI, heritability), their method outperforms various baselines that also rely on deep learning.
Moreover, they observe that several of the genes identified by the GWAS on their image features are known in the literature to have associations with phenotypes related to brain structure or function.

Another experiment shows that the extracted features perform on par with several alternative representations for classifying stroke, cancer and sex on the UKBioBank.

**Audience:**

Yes

**Claims And Evidence:**

Yes

**Requested Changes:**

## Critical

The authors mention that "typical GWAS studies are focused on well-established phenotypes, typically the risks of diseases".
It is easy to imagine why identifying associations between genetic variants and risks of diseases can be useful, eg to monitor more closely individuals at a higher risk.
However, for a reader (such as myself) who is not familiar with GWAS or genomics in general, it is less obvious why it is important to identify associations between genetic variants and abstract, high-dimensional features extracted from brain images with deep learning.
Extracting features as representations for downstream prediction tasks is useful but the authors point out that here it is not the primary goal.
How are the resulting features and associations to be interpreted and used, what are the potential applications or insights?
I think providing more context and motivation in the introduction would be very useful.

The authors mention that non-learning features have limited expressiveness and that dimensionality reduction such as PCA, ICA, NMF are less capable of identifying complex traits.
To support these observations, could the authors add examples of these approaches to the baselines?

Relating those features to other phenotypes relevant to psychology or clinical practice is interesting, and for that the downstream classification task in 6.2 is very interesting.
However the paper does not provide enough information about the training and evaluation procedure to interpret the results in Table 4.
Could the authors add more details to that section?

## Minor

For the final GWAS, a PCA is applied to the learned features and the first 10 are kept.
Does this limit the expressiveness of those features, could the authors explain why it is not a limitation and how the number 10 was chosen.

One of the metrics, the MI estimated with JSE, is exactly what the method is designed to maximize.
It is therefore somewhat expected that it scores higher on that metric.
I do not think this is a limitation especially since other metrics are included but I would suggest including a short comment about this.

**Strengths And Weaknesses:**

## Strengths

The paper is well written.
It clearly states the considered problem, motivates and explains the chosen approach, and describes the experiments and results.
Many baselines are included in the experiments and the proposed method outperforms them on the considered metrics.
An additional experiment evaluates the usefulness of the learned representations for relating imaging features to other phenotypes -- stroke, cancer and sex.

## Weaknesses

I think the motivation for tackling this problem should be stated in more detail.
The baselines only contain deep learning approaches.
I believe including simple baselines that rely on "region-based volumetric or surface features" and linear dimensionality reduction would be helpful to situate the reported metrics.
Too few details are included about the procedure for the "downstream classification performance" evaluation.

---

> ### Author Response · Authors · 2024-02-27
> **Author's response**
>
> Dear Reviewer,
>
> Thank you for your valuable feedback and suggestions. We have revised our manuscript accordingly and below is our itemized response.
>
>
> > The authors mention that "typical GWAS studies are focused on well-established phenotypes, typically the risks of diseases".... I think providing more context and motivation in the introduction would be very useful.
>
> Thank you for the suggestion. We have revised the first paragraph as following to highlight the importance and provide more context:
>
> Genome-wide association studies (GWAS) have been an effective approach driving genetic discovery in the past 15 years~\citep{abdellaoui202315}.
> Given a phenotype of interest and a cohort of individuals with both the measurements of the phenotype and the genotypes over markers across the genome, linear or linear mixed models are built to test for the association of each marker to the phenotype and thus pinpoint the gene loci relevant to the phenotype. While the typical GWAS studies are focused on well-established phenotypes, typically the risks of diseases, or well-established macro-level measurements such as heights, BMI, or molecular-level measurements such as protein and metabolomic biomarkers, there is an interest to expand the scope of GWAS to focus on phenotypes that are derived from high-dimensional complex data modality such as medical imaging data, to better understand the biology of the underlying images. Such derived phenotypes are sometimes called endophenotypes as they serve as intermediate between the genetics and some diseases. For example, brain images are often considered endophenotypes of brain-related diseases such as Alzheimer’s disease. However, there is a lack of sophisticated approaches for deriving phenotypes for GWAS. Again, taking brain imaging as an example, existing approaches mostly used traditional non-learning software to derive brain region-based volumetric or surface features. Such methods, while valuable, are reaching a limit in fully capturing phenotype complexity and can narrow the scope of genetic discoveries. In this context, the advent of deep learning techniques offers a promising avenue for expanding the toolkit available for GWAS.
>
> > The expressiveness of non-learning features, linear approaches, and the motivation of deep learning approaches.
>
> Our intention is not to suggest that deep learning methods are superior across all aspects of genetic discovery. Indeed, linear approaches, often incorporating explicit human knowledge, have proven adept at identifying specific genetic associations that might remain undetected by deep learning methods. Our focus on deep learning stems from its potential to uncover complex, non-linear phenotypes and novel genetic associations, a capacity that complements traditional methods by broadening the horizons of genetic research.
>
> This perspective is supported by literature demonstrating the unique contributions of deep learning. For example, Khush Patel et al. [1] develop an Autoencoder for MRI data identified 13 novel genetic loci not found through linear methodologies. Inspired by such findings, our work seeks to enhance the capability of deep learning in genetic discovery, using the Autoencoder as a starting point.
>
> In our experimental setup, we ensure a fair comparison by employing a consistent encoder architecture across all examined approaches. Given the inherent differences between linear and non-linear methods, their direct comparison can be challenging and might lead to misunderstandings. To avoid this confusion, we have revised our introduction section to more clearly articulate the complementary nature of these approaches.
>
> [1] Patel, Khush, et al. "New phenotype discovery method by unsupervised deep representation learning empowers genetic association studies of brain imaging." medRxiv (2022): 2022-12.
>
> > Relating those features to other phenotypes relevant to psychology or clinical practice is interesting, and for that the downstream classification task in 6.2 is very interesting. However the paper does not provide enough information about the training and evaluation procedure to interpret the results in Table 4. Could the authors add more details to that section?
>
> We have added the following description to the section:
>
> We follow \citet{taleb2022contig} to perform linear evaluation protocol and compute their ROC-AUC. Specifically, we obtain MRI embedding from individual MRI encoders. We then freeze all embeddings and perform linear classification upon the embeddings for the three tasks. Since ROC-AUC is computed on the entire validation set, we repeat the classification on 5 random training-validation splits and show their average ROC-AUC score.

---

> ### Author Response · Authors · 2024-02-27
> **Authors' response cont'd**
>
> > For the final GWAS, a PCA is applied to the learned features and the first 10 are kept. Does this limit the expressiveness of those features, could the authors explain why it is not a limitation and how the number 10 was chosen.
>
> We follow previous studies [2, 3] to use the first 10 PCs for GWAS. This is to enable efficient methodology development and comparisons due to the high computational cost at GWAS step.
>
> [2] Kirchler, Matthias, et al. "transferGWAS: GWAS of images using deep transfer learning." Bioinformatics 38.14 (2022): 3621-3628.
>
> [3] Taleb, Aiham, et al. "Contig: Self-supervised multimodal contrastive learning for medical imaging with genetics." Proceedings of the IEEE/CVF Conference on Computer Vision and Pattern Recognition. 2022.
>
> > One of the metrics, the MI estimated with JSE, is exactly what the method is designed to maximize. It is therefore somewhat expected that it scores higher on that metric. I do not think this is a limitation especially since other metrics are included but I would suggest including a short comment about this.
>
> The purpose of including MI in the metrics is to examine the association between MI and the final outcomes; # loci and heritability, not only for our methods but all baselines. This further justifies the proposed idea of maximizing MI.

---

### Review · Reviewer_V9SE · 2024-01-26

**Summary Of Contributions:**

This paper discusses the unique challenges of Genome-Wide Association Studies (GWAS) in the context of high-dimensional imaging phenotype data and proposes a novel framework, Genetic InfoMax (GIM), which extracts low-dimensional representations from high-dimensional medical imaging data. These representations capture as much information about genetic variations as possible, thereby improving the performance of GWAS. GIM includes a regularized MI estimator, approximated by the Jensen-Shannon Estimator (JSE), and a novel genetics-informed transformer. Experiments on human brain 3D MRI data demonstrate the effectiveness of GIM and a significantly improved performance on GWAS.

**Audience:**

Yes

**Broader Impact Concerns:**

The use of genetic data and imaging data in medical research raises concerns related to privacy. But in the paper there is no discussion on how to address this.

**Claims And Evidence:**

Yes

**Requested Changes:**

This paper contains some language and mathematical typos and mistakes. Below are some examples:
Section 1：
"linear or linear mixed model" should be "linear or linear mixed models".
"and thus limited" should be "thus limiting".

Section 2:
The sub-sentence "the GWAS process involves statistical tests on the sample pairs" should be started as a new sentence; otherwise, the whole sentence is not grammatically correct (comma splice or a run-on sentence).
"each gi \in {0, 1, 2}, representing" should be "and each gi \in {0, 1, 2} represents".
"is formulated as to learn" should be "is formulated as learning".
"two neighbor SNPs gi and gi+1 are not necessarily to be consecutive" should be "two neighboring SNPs, gi and gi+1, are not necessarily consecutive".

Section 3:
"two variables" should be "two sets of variables" or "two multivariates".
"also presents" should be "also arises" or "is also present".
"is formulated as to learn" should be "is formulated as learning".
"both the information of fθ(Y ) and the mutual information I(fθ(Y ),G) is limited" should be "... are limited".
Should "sufficient means of augmentations" be "insufficient means of augmentations"?
"in the right-hand side (RHS) of Eq. (2)" should be "on the right-hand side (RHS) of Eq. (2)".

Section 4：
Line 1 below Eq. (7): "denote" should be "denotes".
Line 1 below Eq. (9): "ranging from 0 and a preset threshold" should be "ranging from 0 to a preset threshold" or "ranging between 0 and a preset threshold".

Section 6.1：
"transmodal contrastive approaches that involves genetic data" should be "transmodal contrastive approaches that involve genetic data".
"The baselines include existing or straightforward training schemes" should be followed by a semicolon (:).

Table 2, column "Change": "+5" should be "+3".

**Strengths And Weaknesses:**

Strengths:
Motivation and Background: The paper thoroughly discusses the unique challenges posed by GWAS and identifies the limitations of contrastive losses in GWAS representation learning. It reviews related work and elaborates on the distinctions between this work and previous research, which is useful for advancing understanding.
Methodological novelty: The paper proposes a novel and effective trans-modal learning framework for addressing the challenges of GWAS in high-dimensional medical imaging data.

Weaknesses:
The clarity of presentation can be improved. Some concepts could be explained in a clearer way.
The use of English needs improvement and should be proofread by a native English speaker. In particular, there are a lot of language and mathematical typos and mistakes.

---

> ### Author Response · Authors · 2024-02-27
> **Authors' response**
>
> Dear Reviewer,
>
> Thank you for your valuable feedback and suggestions. We have revised our manuscript accordingly and below is our itemized response.
>
> > Interpretability and Generalizability
>
> To demonstrate the generalizability to diverse populations, we deliberately split our dataset into a training set to learn representations and a discovery set to run GWAS based on ethnicity. We include a more detailed description on how the dataset is split in Appendix A.
>
> The interpretability is indeed important and is a further step towards understanding the association between gene and brain phenotypes. Interpretability is a future direction of this work. In fact, the learning of informative representations with “good” enough encoders is a crucial prerequisite of interpretability. Once such an encoder model becomes available, we are able to adopt existing visual interpretation or voxel-level explanation approaches [1] to identify significant regions in the brain that are associated with certain genes. We have included the discussion in the future direction part.
>
> [1] Adebayo, Julius, et al. "Sanity checks for saliency maps." Advances in neural information processing systems 31 (2018).
>
> [2] Haar, Lynn Vonder, Timothy Elvira, and Omar Ochoa. "An analysis of explainability methods for convolutional neural networks." Engineering Applications of Artificial Intelligence 117 (2023): 105606.
>
> > Future Directions and Limitations
>
> We have added the following section to discussion limitations and future directions:
>
> Current GWAS relies on statistical testing based on linear models to identify significant loci and compute the heritability scores. However, these methods can be limited in their ability to capture non-linear and complex associations, leading to missed loci and inconsistencies between mutual information and other metrics. To address these limitations, new approaches to discovering non-linear associations are needed.
>
> Regarding mutual information optimization, our current mutual information optimization solution aims to improve representations by capturing more generalizable associations, but it is still limited in reducing information on non-generalizable ones due to a lack of knowledge about non-generalizable patterns. With specific domain knowledge, it is possible to further improve these representations.
>
> The interpretability is a further step towards understanding the association between gene and brain phenotypes, and is another important future direction of this work. In fact, the learning of informative representations with “good” enough encoders is a crucial prerequisite of interpretability. Once such an encoder model becomes available, we are able to adopt existing visual interpretation or voxel-level explanation approaches to identify significant regions in the brain that are associated with certain genes. The interpretability on the genetic encoder side is equally important. While capturing some genetic information, it lacks a resolution to trace back the contributions from individual SNP markers. Future work is needed to develop a more efficient and high resolution genetic encoder.
>
> > Requested changes on grammar errors
>
> We greatly appreciate the thorough examination to improve the readability of our manuscript. We have revised the correspondence.
>
> > Broader Impact Concerns
>
> Thank you for bringing the concern to our attention. We have added the following section:
>
> The use of individualized genetic data and imaging data in medical research raises ethical concerns related to privacy, informed consent, and potential discrimination. The data used in this study were from UK Biobank under the approved project xxxxx\footnote{Concealed for double-blind review purpose} We received ethical approval for this work under xxx-xxx-xx-xxxx from xxx\footnote{Concealed for double-blind review purpose} and followed the best practice and confirm that all necessary patient/participant consent has been obtained and the appropriate institutional forms have been archived, and that any patient/participant/sample identifiers included were not known to anyone (e.g., hospital staff, patients or participants themselves) outside the research group so cannot be used to identify individuals. Participants enrolled in the UK Biobank have provided broad consent, expressly permitting the use of their data for biomedical research purposes that encompass the objectives of this study, ensuring compliance with ethical standards and the intended broad impact of our research.
>
> (1 more paragraph in the revised manuscript)

---

> ### Comment · Reviewer_V9SE · 2024-03-29
> **The revised manuscript demonstrates significant improvements in quality compared to the original version.**
>
> It's encouraging to see that the revisions have addressed most of my concerns raised in the first round of reviews. The paper appears to be in good shape for publication.
>
> Minor edits:
> Page 2: Please insert the conjunction 'and' between the two subsentences in "Here L is the number of SNPs, each gi \in {0, 1, 2} represents the number of carried variants for each individual."
> Page 3, line 1, "two neighbor SNPs":  'neighbor' should be 'neighboring'.
> Page 5, "sufficient means of augmentations": 'sufficient' should be 'insufficient'.
> Section 6.1: The punctuation after "The baselines include existing or straightforward training schemes" should be a comma, not a semicolon.

---

### Review · Reviewer_iD54 · 2024-02-13

**Summary Of Contributions:**

The paper proposes a mutual information inspired framework for the mapping of 3D imaging data to GWAS, by use of a loss function that provides a proxy for mutual information optimization being composed of two losses, i.e. a discriminator loss as well as an autoencoder (AE) reconstruction loss with a regularization term \lambda trading of its impact. The paper further incorporates a transformer based framework for the efficient learning of relevant contexts.

Specifically, the paper considers the Jensen-Shannon Estimator (JSE) to map high-dimensional traits to optimally correspond to genetic profiles of individuals which is related to the noise-contrastive estimation (NCE) optimization in which a discriminator should optimally differentiate corresponding pairs from non-corresponding pairs. However, such loss is found to potentially degenerate focusing on undesirable non-generalizable properties (i.e., referred to as dimensional collapse) which is proposed resolved using the inclusion of the AE loss.

The approach is compared to existing frameworks considering a large UK Biobank datasets of structural MRI and GWAS finding that the approach provides favorable loci identification and downstream task predictions of stroke, cancer and gender in general outperforming existing approaches.

**Audience:**

Yes

**Broader Impact Concerns:**

Broader impact is only briefly discussed including appendix F. It would be good here to expand upon this in terms of how the use of the developed modeling procedures can impact understanding of GWAS and its relation to traits as well as potentially lead to misinterpretations if used incorrectly etc.

**Claims And Evidence:**

No

**Requested Changes:**

The work can be substantially improved in particular by providing:

Improved description of the relation to the regularization by the AE and mutual information – present description below equation 5 are found unclear and insufficient.

Inclusion of error bars on all results for statistical assessment of significance of the results.

Careful description of how to tune \lambda and how the value of \lambda was assessed in the experimentation.

Systematic contrasting the procedure to ContIG. ContIG also uses contrastive loss – is this identical to the considered InfoNCE – to me these appears to be the same please clarify. It would be good to further evaluate how the ContIG contrastive loss compare to the present JSE loss in the experimentation and if JSE change performance compared to the contrastive loss considered in the context of ContIG. The paper should thus compare training with this loss in the results as opposed to the JSE loss to highlight the benefit if any of JSE.

Inclusion of the model ablations in Table 2 also in the downstream task experimentation provided in Table 4 regarding classification of stroke, cancer and gender.

**Strengths And Weaknesses:**

Strengths

•	The paper is in general well written.

•	The paper appear to improve upon current state-of-the-art for relating high-dimensional traits (structural MRI) to GWAS.

•	The dataset is used is large (UK Biobank).

•	The approach appears sound.

Weaknesses

•	Complete lack of error bars in the assessments making it very hard to judge the significance of the approach when compared to baselines (I therefore rate claim and evidence insufficient).

•	The derivation although strongly grounded in mutual information ends up being a rather simple standard loss function trading of auto-encoder reconstruction and discriminator losses with rather limited novelty here.

•	Transformers are well known to provide improved context learning and the inclusion of such architecture is also of limited novelty.

•	How to tune the regularization \lambda is very unclear and it is not discussed how it influences results nor how the tuning of \lambda is practically done in the experimentation.

Additionally, I find the relation between entropy and least squares reconstruction regularization to not be well elaborated and unclear in the presentation although this is part of the central contributions apart from the proposed use of transformers. I.e., what is H(f_theta(Y)|Y) and what does the crossing arrow and zero here mean (this term goes to zero?).  I assume I(f_theta(Y),Y) is the mutual information but how it that estimated here (is it using the  JSE og NCE estimators?) how exactly this relates to the reconstruction regularization should further be sharpened as the text below equation 6 is unclear and could be better elaborated upon.

Minor comment:
Is there a mistake in table 2.1 in which Change of +5 should be +3?
Table 3 and 4 does the autoencoder correspond to setting \lambda->infty?

In general this is a well written paper, however, I find the technical contribution rather limited and the results unconvincing due to the lack of error bars. I.e., how do results change according to random initialization (and potentially changes to training and test splits)? Such uncertainty assessment is necessary to adequately judge the current results.

---

> ### Author Response · Authors · 2024-02-27
> **Authors' response**
>
> Dear Reviewer,
>
> Thank you for your valuable feedbacks and suggestions. We have revised our manuscript accordingly and below is our itemized response.
>
> > Improved description of the relation to the regularization by the AE and mutual information – present description below equation 5 are found unclear and insufficient.
>
> In Equation 5, we use the reconstruction loss as a proxy to maximize the entropy of f(Y) as the entropy is not directly available. In Equation 6, we justify the proxy by showing the equivalence between H(f(Y)) and the mutual information I(f(Y), Y), because the conditional entropy H(f(Y)|Y) reduces to zero when the neural network f is deterministic. Intuitively, given the input Y, the learned embedding f(Y) will not carry additional information about Y.
>
> We then consider the maximization of the mutual information I(f(Y), Y). The objective consists of two components: 1) the alignment of f(Y) to Y, which is achieved by maximize the likelihood that f(Y) and Y in a positive pair belong to the same individual, and 2) the uniformity of f(Y), which is achieved by maximizing the likelihood that f(Y) and any other Y’ are independent. To reduce the batch-size sensitivity and optimize memory cost for extremely high-dimensional MRI data, we avoid introducing additional encoders and discriminator to use typical MI estimators. Instead, we use the reconstruction term to maximize the log-likelihood that f(Y) and Y in a positive pair belong to the same individual, corresponding to the first component. The uniformity of f(Y) is handled in I(f(Y), G) by maximizing the likelihood that f(Y) and any other G’ are independent. Intuitively, it guarantees that the embeddings of any two different individuals should be distinguishable from each other.
>
> > Inclusion of error bars on all results for statistical assessment of significance of the results.
>
> We have included the error bar to the estimated MI and heritability in both Table 1, Table 2, and Table 4 to demonstrate the statistical significance. For the number of loci, we would like to note that the identification of loci is already based on statistical significance. For table 1, a significance threshold of p-value < 5e-9 is used. The numbers of identified loci for T2 under different thresholds are also compared in Table 3. To demonstrate the robustness of GIM, we have obtained additional results by training 3 additional GIM models independently. They resulted in 43, 42, 46 discovered loci, respectively, which indicates consistent and significant improvement compared to baseline approaches.
>
> > Careful description of how to tune \lambda and how the value of \lambda was assessed in the experimentation.
>
> The GWAS computation is heavy and hypertuning is usually expensive and unaffordable. So we did not perform any fine-grained hyper-tuning but only focused on comparing different frameworks. For \lambda, we use an heuristic value of 1 without exhaustively tuning, as described in Appendix B - Training.
>
> > Systematic contrasting the procedure to ContIG.
>
> ContIG indeed uses InfoNCE loss. And we include the results of ContIG as a baseline in the Trans-Modal Contrastive group, shown as “InfoNCE (ContIG, Taleb et al. (2022))” in Table 1. We exactly follow the work to reproduce the implementation of the objective and configurations, e.g. using the weighted sum of I_{NCE}(X, Y) and I_{NCE}(Y, X), and more details in Appendix B.
>
> Regarding the difference between NCE and JSE, we do not claim any general advantage of using JSE over NCE. Both estimators can be used in our framework. Our intuition of choosing JSE is due to the limited GPU memory for both MRI and genetic encoders and JSE usually has a more robust performance under smaller batch sizes.
>
> > Inclusion of the model ablations in Table 2 also in the downstream task experimentation provided in Table 4 regarding classification of stroke, cancer and gender.
>
> We have updated the results in Table 4 including ablations of GIM, their corresponding error bar, and stronger baseline results by involving fine-tuning for the downstream tasks. For cancer and biological sex classification, we identify continuous performance gain when adding individual components, whereas, for stroke, the regularized loss and more intensive cropping bring the most improvement. For Autoencoder, we observe significantly better performance on biological sex classification. The underperformance of cross-modal approaches is due to the exclusion of the sex chromosome in the given SNP data, and thus the MI maximization discourages the model to capture biological sex-related information from MRI data.

---

> ### Author Response · Authors · 2024-02-27
> **Authors' response cont'd**
>
> > The derivation although strongly grounded in mutual information ends up being a rather simple standard loss function trading of auto-encoder reconstruction and discriminator losses with rather limited novelty here.
>
> Thank you for your valuable feedback. We appreciate the opportunity to clarify the novelty and contributions of our work further. Our research primarily focuses on exploring the limitations of conventional contrastive learning objectives within the context of high-dimensional GWAS. This setting presents unique challenges that have not been touched in existing literature, particularly due to the high-dimensional nature of MRI and the unique form of genetic data.
>
> The core contribution of our work lies in the development of a novel learning framework specifically designed to bridge the gap in applying deep learning methods to GWAS problems. Our work aims to address a unique problem rather than simply introducing a new learning objective. The proposed learning framework, driven by the unique challenges of a novel problem, to our knowledge, has not been proposed or tested in prior studies. In fact, the simplicity of our loss is one of its strengths, allowing for efficient computation without sacrificing effectiveness, thereby making it more accessible for practical applications.
>
> In addition, though simple in formulation, the derivation of the GIM objective is not straightforward and involves a thorough analysis that yields significant insights. This aspect of our work contributes to the broader understanding of how to effectively leverage deep learning for GWAS, a topic that has received limited attention in the past. Our findings and discussions provide valuable perspectives that can inspire future research in this area.
>
> > Transformers are well known to provide improved context learning and the inclusion of such architecture is also of limited novelty.
>
> We acknowledge that transformers have become a cornerstone in advancing context learning across various domains due to their ability to capture long-range dependencies. However, the novelty of our work does not solely lie in utilizing transformer architecture, but the significant adaptation of this architecture to address the unique and complex challenges presented by genetic data, specifically SNP data. Our work introduces a genetics-informed transformer model, a novel adaptation that integrates domain-specific knowledge to enhance the model's ability to discern and leverage the intricate dependencies among genes. To our knowledge, this is the first attempt to design a transformer that is not only informed by but also towards an optimized approach for genetics data.
>
> > Minor comment: Is there a mistake in table 2.1 in which Change of +5 should be +3? Table 3 and 4 does the autoencoder correspond to setting \lambda->infty?
>
> Yes. The change should be +3 instead of +5. And Auto-Encoder is equivalent to setting \lambda to infinity.

---

### Decision · Action_Editor_Uzek · 2024-04-13

**Recommendation:** Accept with minor revision

**Comment:**

Please do a thorough pass on the writing.

**Audience:**

This paper borrows from different field (GWAS, 3D imaging) so that it can be considered by people from many different audiences. Overal, it should be of interest to people working on applications, particularly biomedical applications.

**Claims And Evidence:**

The paper has been found to be sound by the reviewers, though quite incremental.
On the positive side, the authors have better substantiated their claims with additional experiments and better presentation of the results.
On the negative side, the technical novelty is limited.
Could the authors do a last editing effort  (grammar, sentence structure, vocabulary choice, etc.) ? Now there exist many automated tools to help in that.
The consensus among reviewers is that it should be accepted.